# Antimicrobial and Antibiofilm Effect of Bacteriocin-Producing *Pediococcus inopinatus* K35 Isolated from Kimchi against Multidrug-Resistant *Pseudomonas aeruginosa*

**DOI:** 10.3390/antibiotics12040676

**Published:** 2023-03-30

**Authors:** Eun-Ji Yi, Ae-Jung Kim

**Affiliations:** Department of Alternative Medicine, Kyonggi University, Seoul 03746, Republic of Korea

**Keywords:** *Pediococcus inopinatus*, biofilm formation, antimicrobial activity, multidrug-resistant, *Pseudomonas aeruginosa*, kimchi

## Abstract

Background: Recently, the emergence of multidrug-resistant bacteria due to the misuse of antibiotics has attracted attention as a global public health problem. Many studies have found that fermented foods are good sources of probiotics that are beneficial to the human immune system. Therefore, in this study, we tried to find a substance for the safe alternative treatment of multidrug-resistant bacterial infection in kimchi, a traditional fermented food from Korea. Method: Antimicrobial activity and antibiofilm activity were assessed against multidrug-resistant (MDR) *Pseudomonas aeruginosa* using cell-free supernatants of lactic acid bacteria (LAB) isolated from kimchi. Then, UPLC-QTOF-MS analysis was performed to detect the substances responsible for the antimicrobial effect. Results: The cell-free supernatant (CFS) of strain K35 isolated from kimchi effectively inhibited the growth of MDR *P. aeruginosa*. Similarly, CFS from strain K35 combined with *P. aeruginosa* co-cultures produced significant inhibition of biofilm formation upon testing. On the basis of 16s rRNA gene sequence similarity, strain K35 was identified as *Pediococcus inopinatus*. As a result of UPLC-QTOF-MS analysis of the CFS of *P. inopinatus* K35, curacin A and pediocin A were detected. Conclusion: As a result of this study, it was confirmed that *P. inopinatus* isolated from kimchi significantly reduced MDR *P. aeruginosa* growth and biofilm formation. Therefore, kimchi may emerge as a potential source of bacteria able to help manage diseases associated with antibiotic-resistant infections.

## 1. Introduction

*Pseudomonas aeruginosa* is a causative pathogenic bacterium ubiquitously known for its metabolic versatility [1]. Infection with *P. aeruginosa* can range from mild inflammation to lethal systemic disease [2]. Unfortunately, the current guidelines for *P. aeruginosa* are barely effective due to the low permeability of the signature outer membrane of this Gram-negative bacterium. This property of metabolic flexibility makes it prone to resistance to the current antibiotics on the market [3,4]. Given the seriousness of *P. aeruginosa* infections and the limited antimicrobial arsenal available to treat them, new antimicrobials with distinct mechanisms of action are desperately needed [5].

Recently, Shokri et al. reported a mechanism related to biofilm formation as a cause of resistance in *P. aeruginosa*, revealing that 76 (or 95%) of the studied *P. aeruginosa* strains produced biofilms as a defensive mechanism against antibiotics [6]. Hence, the structure of biofilms may prevent antimicrobial drugs from penetrating the bacterium, as some charged inhibitors can bind to the oppositely charged polymers of the biofilm matrix [7,8]. Several pathways are responsible for the resistance provided by biofilms, but the most prevalently known are stated in Figure 1.

To address the previously mentioned issue of controlling multidrug-resistant (MDR) *P. aeruginosa* biofilms, widely used probiotic lactic acid bacteria (LAB) such as *Enterococcus*, *Streptococcus*, *Bifidobacterium*, and *Lactobacillus* species assist in the treatment and control of gut microbiota and have been considered safe choices for the treatment of immune-related diseases such as atopic disease [9]. Various previous studies have indicated that an antibiofilm effect against *P. aeruginosa* strains can be achieved using LAB metabolites (Table 1). As probiotics, LAB must be able to tolerate pancreatic digestive enzymes, bile, and acids, and eventually to adhere to and function within the intestinal epithelial tissue of the colon [10]. Hence, LAB strains could be a safe option in light of the positive effects associated with specific species or genus strains [11].

Bacteriocins are a group of antimicrobial peptides produced by bacteria that have been studied for their potential therapeutic applications. Bacteriocins in LAB are classified into four groups based on their structural and functional properties. These include lantibiotics, which are modified bacteriocins (class I); non-lantibiotics, which are heat-stable and unmodified (class II); a group of large heat-labile bacteriocins (class III); and class IV, which often includes molecules that are intricate and contain lipid and carbohydrate components. Recent studies have focused more on the potential of class II bacteriocins as novel therapeutic agents [16].

Along the various sources of LAB strains listed in Table 1, the Korean traditional food kimchi is known for the use of LAB as its starter organisms [1]. In previous studies, kimchi-derived LAB were shown to have antimicrobial [3], anti-adipogenic [4], and antioxidant [3,5] effects, making kimchi a potential source of LAB for our study on MDR *P. aeruginosa*.

*Pediococcus inopinatus* is a Gram-positive lactic acid bacterial species belonging to the genus *Pediococcus* and the family Lactobacillaceae, which can exist naturally in plants and fruits and is frequently connected to breweries or other settings in which alcoholic beverages are made [12], including wine [13]. The *Pediococcus* genus consists of several species, including *P. acidilactici*, *P. pentosaceus*, *P. damnosus*, *P. parvulus*, *P. inopinatus*, *P. halophilus*, *P. dextrinicus*, and *P. urinaeequi* [17]. Members of the genus *Pediococcus* are facultative anaerobes, non-motile, and non-spore-forming. The antimicrobial properties of *Pediococcus* species, particularly their bacteriocins, have been extensively studied for their potential use in food preservation and as therapeutic agents [16]. Furthermore, the presence of *Pediococcus* species in fermented foods and beverages highlights their importance in promoting gut health and overall wellbeing, since LAB are famous for being safe and are often used as dairy product starters. Therefore, understanding the properties and applications of the genus *Pediococcus* could provide new avenues for developing novel therapies and improving food safety, especially in terms of antibiotic-resistant bacteria like *P. aeruginosa*. *P. inopinatus* was shown to have an anti-allergy effect when used to produce lactobacillus-fermented soybean milk curd. The in vivo study by Kang et al. showned that *P. inopinatus*-fermented soy curd had much better anti-allergy properties than soy milk alone for atopic dermatitis treatment [14]. While few studies have been conducted on this LAB strain, it could be a potential option for MDR *P. aeruginosa* studies.

In this study, we isolated LAB, which are known to have beneficial effects on the human immune system, from traditional fermented foods and evaluated their potential as a safe alternative treatment for multidrug-resistant bacterial infections.

## 2. Results

### 2.1. Isolation of Antimicrobial LAB from Kimchi

Strains were isolated from kimchi using the color-changing characteristic of bromocresol purple (BCP), which changes from purple to yellow when the pH is lowered by LAB. In this study, 40 strains that formed yellow rings around colonies on BCP agar (Eiken Chemical Co., Ltd., Tokyo, Japan) plates were selected and used for antimicrobial screening. As a result of the antimicrobial screening of the 40 selected strains, according to the disc diffusion assay, strain K35 showed antimicrobial activity against *P. aeruginosa* and was selected. The zones of inhibition of antimicrobial activity were 17.5 ± 0.87, 17.33 ± 1.26, and 18.5 ± 0.50 mm against *P. aeruginosa* KCTC 2513, *P. aeruginosa* CCARM 0223, and *P. aeruginosa* CCARM 0224, respectively (Figure 2 and Table 2), and the minimum inhibitory concentration (MIC) and minimal bactericidal concentration (MBC) values were 2.5 and 5 mg/mL against *P. aeruginosa* KCTC 2513, *P. aeruginosa* CCARM 0223, and *P. aeruginosa* CCARM 0224 (Table 3).

### 2.2. Identification of Strain K35

Phylogenetic analysis was conducted using the 16s rRNA gene. The sequence similarity of strain K35 indicated that its closest relative was *Pediococcus inopinatus* DSM 20285^T^ (99.79%). This relationship between strain K35 and other members of the genus *Pediococcus* was also evident in the phylogenetic tree (Figure 3). Biochemical and cultural characteristics were analyzed using *Pediococcus inopinatus* DSM 20285^T^, which had the highest similarity according to the phylogenetic analysis, as a comparison strain (Table 4). In the growth temperature test, both strains were able to grow at 25 to 45 °C, and the most suitable temperature was 30 °C. In API 50CH tests, both strains were positive for acid production from D-galactose, D-glucose, D-fructose, D-mannose, N-acetylglucosamine, amygdalin, esculin ferric citrate, salicin, D-cellobiose, D-maltose, D-trehalose, and gentiobiose, while both strains were negative for acid production from glycerol, erythritol, D-arabinose, L-arabinose, D-ribose, L-xylose, D-xylose, D-adonitol, methyl-β-D-xyopyranoside, L-sorbose, L-rhamnose, dulcitol, inositol, D-mannitol, D-sorbitol, methyl-α-D-glucopyranoside, arbutin, D-lactose, D-melibiose, D-saccharose, inulin, D-melezitose, D-raffinose, amidon, glycogen, xylitol, D-lyxose, D-tagatose, D-fucose, L-fucose, D-arabitol, L-arabitol, potassium gluconate, potassium 2-ketogluconate, and potassium 5-ketogluconate.

### 2.3. Biofilm Formation Inhibitory Activity

The results of the test of biofilm formation inhibitory effect against *P. aeruginosa* are shown in Figure 4. In the control group, it was shown that biofilm formation increased in a concentration-dependent manner for *P. aeruginosa* KCTC 2513 and *P. aeruginosa* CCARM 0224, but for *P. aeruginosa* CCARM 0223, the change in biofilm amount according to the concentration was not significant. In the K35 treatment group, evaporated CFS did not affect the biofilm formation of *P. aeruginosa* strains below the MIC concentration, but biofilm formation was significantly inhibited at concentrations above the MIC. It is assumed that the MRS broth (Difco, Detroit, MI, USA) contained in the evaporated CFS of K35 promoted the growth of *P. aeruginosa*, but at concentrations above the MIC, both growth and biofilm formation were inhibited by the metabolites of K35.

### 2.4. Observation of the Morphological Changes of Bacterial Cells

The morphological changes of *P. aeruginosa* KCTC 2513, *P. aeruginosa* CCARM 0223, and *P. aeruginosa* CCARM 0224 following treatment with the MBC of CFS from K35 were visualized using scanning electron microscopy (SEM) (Figure 5). Cell surface damage and shortened cell length were observed in the treatment group compared to the control group. In particular, in *P. aeruginosa* CCARM 0224, evaporated CFS treatment resulted in complete deterioration and cell debris was observed.

### 2.5. Detection of Substances with Antimicrobial and Anti-Biofilm-Formation Effects

The total ion chromatogram (TIC) and the UV chromatograms of the K35 culture broth are shown in Figure 6. The tentative identification of compounds was performed using Waters software (UNIFI v1.8.1) and an online database (ChemSpider). Compound identification was based on molecular ion mass and fragmentation pattern, and, as a result, curacin A and pediocin A, known as antimicrobial compounds, were tentatively identified in K35. Table 5 shows the elution time, molecular formula, molecular weight, and mass error of the tentatively identified compounds.

## 3. Discussion

In this study, we attempted to demonstrate that the CFS of *P. inopinatus* K35 isolated from kimchi has antimicrobial activity and anti-biofilm-formation activity against MDR *P. aeruginosa*. Recently, as the number of antibiotic-resistant pathogens has greatly increased, the need for research to find new alternative treatments to replace existing antibiotic treatments without serious side effects or toxicity is emerging [18,19]. For this reason, studies on the possible application of LAB metabolites that are effective for the inhibition of *P. aeruginosa* infection are increasing [6,12,13,14,15].

Several studies have reported the anti-allergic and antimicrobial activity of *P. inopinatus* [20], and this bacterium has also been shown to inhibit the growth of *Escherichia coli*, *Helicobacter pylori*, *Listeria monocytogenes*, and *Staphylococcus aureus* [21]. However, data on resistant strains have still not been reported.

According to information provided by the National Research Institute of Korea, *P. aeruginosa* CCARM 0223 and *P. aeruginosa* CCARM 0224 strains are resistant to ampicillin, cephalothin, gentamicin, and norfloxacin. In this study, the CFS of *P. inopinatus* K35 significantly reduced the growth and biofilm formation of *P. aeruginosa* CCARM 0223 and *P. aeruginosa* CCARM 0224, which complements the other reports on the antibiofilm effect of LAB that were summarized in Table 1.

Previous studies have reported that antibiotic-resistant strains have the tendency to display enhanced biofilm formation, making it more difficult for antibiotics to penetrate [15]. This biofilm strengthening is consistent with the results of this study, in which the antibiotic-resistant strains *P. aeruginosa* CCARM 0223 and *P. aeruginosa* CCARM0224 produced more biofilm than the normal strain *P. aeruginosa* KCTC 2513 (Figure 4).

The results of our study confirm that even in the face of the strong biofilm formation ability exhibited by antibiotic-resistant bacteria, the cell-free supernatant of K35 can effectively inhibit biofilm formation in MDR *P. aeruginosa* strains at concentrations of 2.5 mg/mL or higher, as shown in Figure 4. Therefore, these findings suggest that the lactic acid bacteria isolated in this study have the potential to serve as a viable alternative for inhibition of antibiotic-resistant bacteria.

A recent research work stated that antibiotic-resistant strains were observed to have greater cell elongation than normal bacteria [15]. Another study reported that antibiotics such as erythromycin and clarithromycin inhibit the protein synthesis of *P. aeruginosa* and eventually result in the loss of cell viability [22]. The shrinkage of the *P. aeruginosa* cells in the treated samples in our work can be attributed to the possible ability of the CFS of K35 to inhibit protein synthesis (Figure 5).

Based on previous findings that antibiotics inhibit pathogens by interfering with the normal cycle of maintenance of cell integrity and by causing cell shrinkage and loss of cell viability [23,24], it is assumed that the CFS of *P. inopinatus* K35 effectively penetrates the cell wall of *P. aeruginosa*, interferes with cell growth and normal cell activity, and ultimately inhibits biofilm formation. The efficacy of *P. inopinatus* K35 was the strongest against *P. aeruginosa* CCARM 0224, in which it was shown to lead to complete deterioration.

In the context of lactic acid bacteria from food, antimicrobial activity is generally attributed to the presence of organic acids, hydrogen peroxide, or bacteriocins [25,26]. To determine the origin of the antimicrobial activity displayed by the studied *P. inopinatus* K35, we employed UPLC-TOF-MS analysis. In the chromatography results, curacin A and pediocin A were tentatively identified in *P. inopinatus* K35 cultured broth.

While curacin A has demonstrated potent antiproliferative and antimitotic effects against human cancer cells [27], its effectiveness as an antimicrobial agent against MDR *P. aeruginosa* remains to be determined. The differences between prokaryotic (such as *P. aeruginosa*) and eukaryotic (such as human cancer) cells extend beyond merely their structures; the membrane composition and genetic material also differ significantly [28]. While eukaryotes have a membrane-bound nucleus and other membrane-bound organelles, prokaryotes lack these. Hence, further research is essential to evaluate the potential and to clarify the detailed mechanism of curacin A as an antimicrobial agent against pathogenic bacteria in general and specifically against MDR *P. aeruginosa*. Pediocin A, on the other hand, is mainly found in *Pediococcus* spp. and is known to inhibit pathogens such as *Bacillus cereus* and *Listeria monocytogenes* effectively [25,26]. These two small antimicrobial peptides, classified as class II bacteriocins, have the potential to be highly specific in their targeting and have a lower likelihood of causing the development of resistance compared to traditional antibiotics [29]. Therefore, these findings suggest that the inhibition of growth and biofilm formation of *P. aeruginosa* by *P. inopinatus* K35 may be due to curacin A and pediocin A, representing a promising step toward the development of new and effective treatments for this difficult-to-treat infection. However, further research is needed to understand their mechanisms of action and to evaluate their efficacy and safety in clinical settings.

## 4. Conclusions

In this study, we isolated *P. inopinatus* K35 from kimchi, confirmed its antimicrobial activity, and confirmed that this strain effectively inhibits the growth and biofilm formation of three *P. aeruginosa* strains. In addition, through metabolomic analysis, *P. inopinatus* K35 was tentatively shown to produce curacin A and pediocin A, and it is suggested that it exerts antibacterial activity against multidrug-resistant bacteria through the production of these bacteriocins. These findings indicate that the lactic acid bacterium K35 isolated from kimchi presents the potential to be developed as an alternative to antibiotics to treat infections caused by *P. aeruginosa*. It is also suggested that kimchi can be a good source for the isolation of useful microorganisms with antimicrobial ability against multidrug-resistant bacteria.

## 5. Materials and Methods

### 5.1. Isolation and Identification of LAB from Kimchi

Kimchi soup was serially diluted in 0.85% (*w*/*v*) saline solution to a concentration of 10^−6^ to 10^−9^, and was spread onto BCP agar. The plates were then incubated for 2 days at 30 °C. Single colonies that displayed yellow rings around them on BCP plates were transferred to MRS agar plates and routinely cultured on MRS agar and in MRS broth at 30 °C.

To identify the homological strains of the isolates, the 16s rRNA sequence technique with primers 1492R (5′ TACGGYTACCTTGTTACGACTT 3′) and 27F (5′ AGAGTTTGATCCTGGCTCAG 3′) was used. Then, the EzBioCloud database (https://www.ezbiocloud.net/identify, accessed on 15 November 2022) was used to analyze the 16s rRNA sequences. A phylogenetic tree was reconstructed using the neighbor-joining method and the maximum-parsimony method in the MEGA X program with bootstrap values based on 1000 replications [30,31]. For biochemical analysis, a carbohydrate fermentation test was performed using an analytical profile index (API) 50 CH kit (BioMérieux, Marcy l’Etoile, France) [32].

### 5.2. Bacterial Strains and Growth Conditions

*P. aeruginosa* KCTC 2513, *P. aeruginosa* CCARM 0223, and *P. aeruginosa* CCARM 0224 were purchased from the Korean Collection for Type Cultures (KCTC). *P. aeruginosa* strains were cultured in nutrient broth (NB; Difco, Detroit, MI, USA) agar under aerobic conditions at 30 °C. Then, the selected LAB was grown in MRS broth for 48 h at 30 °C to produce a mass amount of metabolites. Culture medium was then centrifuged for 10 min at 12,000 rpm, and the supernatant was filtered using a 0.22 μm membrane filter. Subsequently, the filtrates were used directly as samples for antimicrobial activity screening, and filtrates concentrated using a centrifugal evaporator (EYELA, Tokyo, Japan) at 45 °C were utilized for further antimicrobial and antibiofilm tests.

### 5.3. Disc Diffusion Assay

LAB with antimicrobial properties were identified using the standard disc diffusion approach following the method of Zaidan et al. [33]. First, 100 µL of filtered supernatant was applied to sterile filter paper discs (8 mm diameter, Whatman No. 1), then the discs were placed on Mueller–Hinton agar (MHA; Difco, Detroit, MI, USA) plates spread with pathogen indicators (1 × 10^6^ UFC/mL) and incubated at 30 °C for 24 h. The antimicrobial effects of LAB were analyzed by measuring the diameter of the inhibitory zone in millimeters using calipers [34].

### 5.4. Broth Microdilution Method

The broth microdilution method described in the Clinical and Laboratory Standards Institute guidelines was utilized to confirm the MIC and MBC of the selected LAB. A 96-well microtiter plate (Thermo Fisher Scientific, Waltham, MA, USA) was used to inoculate serially diluted evaporated CFS of the selected LAB. To each well containing 100 μL of serially diluted evaporated CFS, 100 μL of *P. aeruginosa* strains in NB (1 × 10^6^ CFU/mL) was added, and the plates were incubated at 30 °C for 24 h. Then, the optical density (OD) was measured at 595 nm using a microplate reader (Molecular Devices, San Francisco, CA, USA), and the concentration showing an OD of 20% or less compared to the control group was set as the MIC. Subsequently, a portion of each well was streaked onto a NB agar plate using a 1 uL inoculation loop. The plates were incubated at 37 °C for 24 h and the number of colonies was observed. The lowest concentration without colony growth on the NB agar plates was recorded as the MBC.

### 5.5. Biofilm Formation Inhibition Assay

To determine whether treatment of evaporated CFS reduced the amount of biofilm, 100 μL each of the diluted *P. aeruginosa* strains (1 × 10^6^ CFU/mL) and different concentrations of CFS-K35 (0, 0.31, 0.63, 1.25, 2.5 and 5 mg/mL) were added into 96-well microtiter plates and incubated for 24 h at 37 °C without shaking. After incubation, the solution was discarded, and each well was washed with phosphate-buffered saline (PBS) twice. Subsequently, 100 µL of 0.01% crystal violet solution (in 0.1% acetic acid and 95% ethanol) was added to stain each well for 15 min, followed by rinsing twice with PBS to remove non-specific staining and air drying at room temperature. Fixed crystal violet was released using 33% acetic acid, and then the biofilm formation inhibition was evaluated by measuring OD at 595 nm using a microplate reader. Evaporated MRS broth was used as a control during the biofilm formation test.

### 5.6. Scanning Electron Microscopy (SEM) of Bacteria

Dilutions of 1 × 10^7^ CFU/mL of *P. aeruginosa* were reacted with the MBC of evaporated CFS at 37 °C for 24 h. *P. aeruginosa* samples were then fixed for 3 h at 4 °C with 2.5% glutaraldehyde in PBS. The fixed samples were dehydrated using ethanol solutions with progressively higher concentrations (30%, 50%, 70%, 80%, 90%, and 100%). After dehydration, the bacteria were dried overnight by adding 100 μL of hexamethyldisilazane (Sigma Aldrich, St. Louis, MO, USA). Carbon tape was attached to the stub and dried samples were mounted on it. Then, plasma was generated on the surface of the non-conductive sample to create a metal film. Bacteria were imaged using a SU8010 scanning electron microscope (Hitachi, Tokyo, Japan).

### 5.7. UPLC-QTOF-MS Analysis

Further analysis was conducted using an UPLC (ACQUITY UPLC I-Class System, Waters, Milford, MA, USA) connected to a quadrupole time-of-flight (Q-TOF) mass spectrometer detector (Xevo G2–XS, Waters, Milford, MA, USA). The mass spectrometer was operated in negative-ion electrospray mode with a mobility-enabled non-targeted HDMSE scan method, with a range of 50–2000 Da and a 0.1 s scan time. In this investigation, 0.1% formic acid in water and acetonitrile were used as the mobile phases A and B, respectively, to produce chromatographic separation at a flow rate of 0.35 mL/min. The elution conditions were as follows: 0–1 min, 0–1 min, 0% B; 1–4 min, 0–55% B; 4–14 min, 55–90% B; 14–14.5 min, 90–99% B; 14.5–17 min, 99% B; 17–17.1 min 99–0% B; 17.1–20 min 0% B. Detection wavelengths of 254 and 320 nm were used. The energies of the MS low and high collisions were 6 and 20 to 60 eV, respectively. The drying gas utilized was nitrogen. The cone gas flow was kept at 50 L/h, while the desolvation gas flow was 800 L/h. The source temperature was 100 °C, whereas the desolvation temperature was 350 °C. The capillary and sampling cones were 1.8 KV and 30 V. A 2.0 KV reference capillary was used. Data were gathered and processed using UNIFI v1.8.1 software.

### 5.8. Statistical Analysis

GraphPad Prism 9 (GraphPad Software Inc., La Jolla, CA, USA) was utilized to analyze the one-way ANOVA and two-way ANOVA. The data in this study were collected in three replications each and are presented as mean ± standard deviation. *p* < 0.05 was regarded as a significant value in all statistical analyses.

## Figures and Tables

**Figure 1 antibiotics-12-00676-f001:**
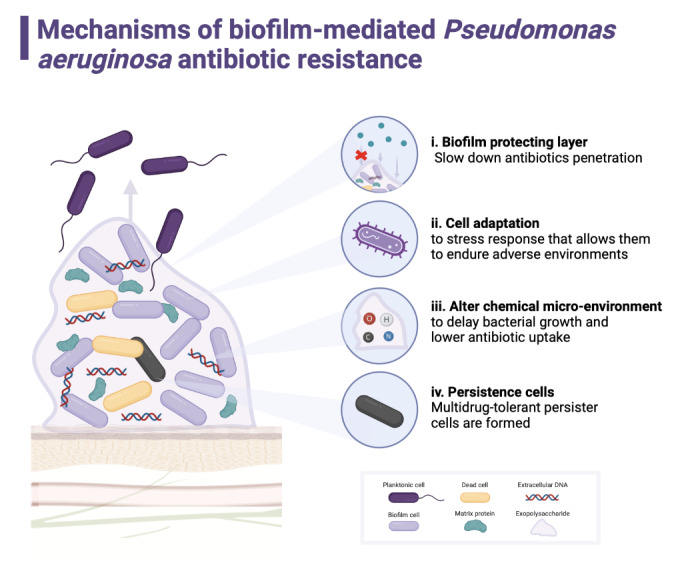
Mechanisms of *P. aeruginosa* biofilm-mediated antibiotic resistance.

**Figure 2 antibiotics-12-00676-f002:**
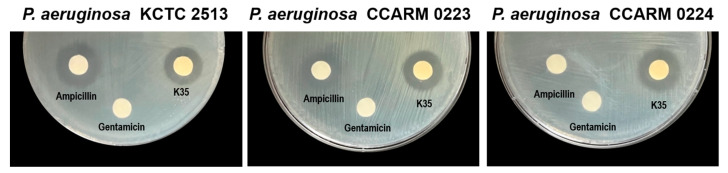
Antimicrobial activity by CFS of *P. inopinatus* K35 against *P. aeruginosa* KCTC 2513, CCARM 0223, and CCARM 0224.

**Figure 3 antibiotics-12-00676-f003:**
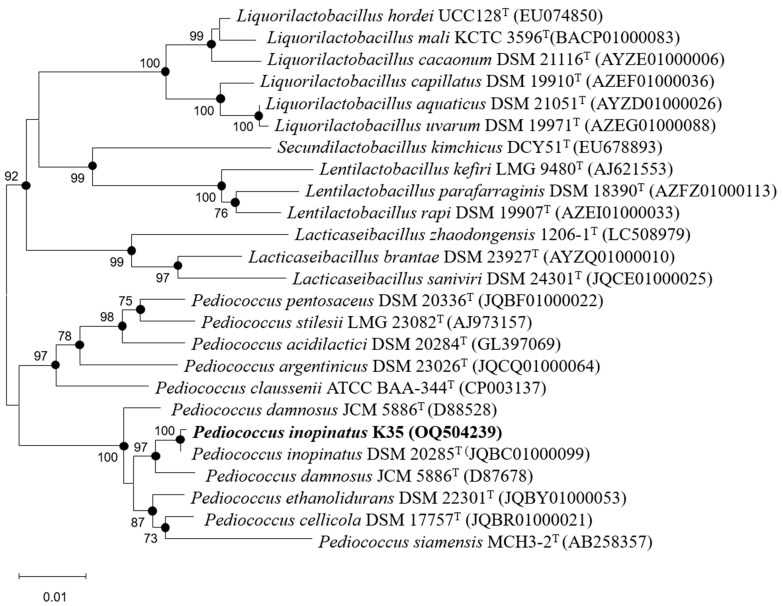
Neighbor-joining phylogenetic tree of strain K35. Bootstrap values (expressed as a percentage of 1000 replications) > 65% are shown at the branch points. Bar, 0.01 substitutions per nucleotide position.

**Figure 4 antibiotics-12-00676-f004:**
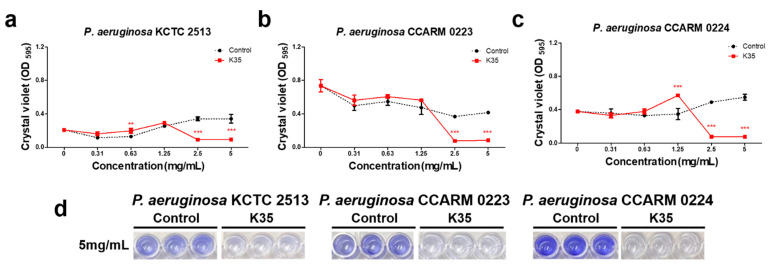
Anti-biofilm-formation activities of *P. inopinatus* K35 against (**a**) *P. aeruginosa* KCTC 2513, (**b**) *P. aeruginosa* CCARM 0223, and (**c**) *P. aeruginosa* CCARM 0224. (**d**) Representative images of biofilm inhibition by *P. inopinatus* K35 are shown. Data are presented as mean ± SD of the results from three replicates. ** *p* ≤ 0.01, *** *p* ≤ 0.001.

**Figure 5 antibiotics-12-00676-f005:**
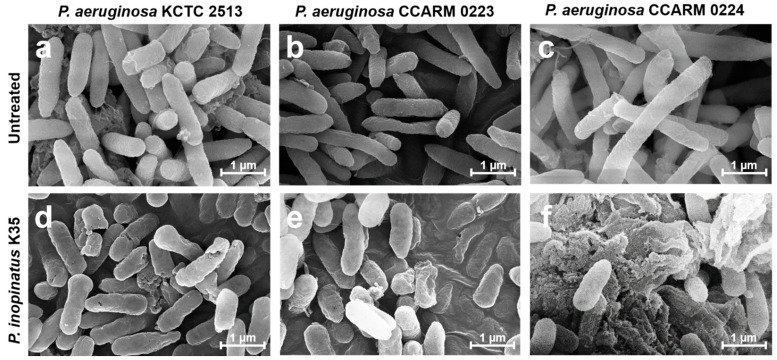
Scanning electron microscopy (SEM) (magnification, 50,000×, bar 1 µm) images of *P. aeruginosa.* (**a**) Untreated *P. aeruginosa* KCTC 2513 cells. (**b**) Untreated *P. aeruginosa* CCARM 0223 cells. (**c**) Untreated *P. aeruginosa* CCARM 0224 cells. (**d**) *P. aeruginosa* KCTC 2513 cells treated with 5 mg/mL of *P. inopinatus* K35 CFS for 24 h. (**e**) *P. aeruginosa* CCARM 0223 cells treated with 5 mg/mL of *P. inopinatus* K35 CFS for 24 h. (**f**) *P. aeruginosa* CCARM 0224 cells treated with 5 mg/mL of *P. inopinatus* K35 CFS for 24 h.

**Figure 6 antibiotics-12-00676-f006:**
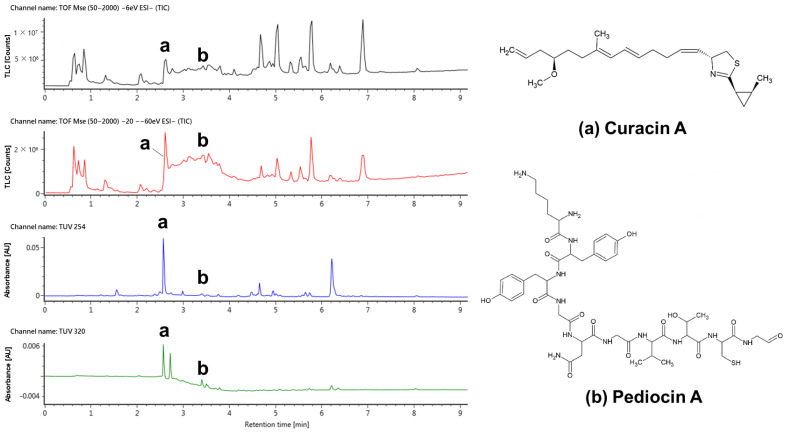
UPLC-QTOF-MS chromatogram of cell-free supernatant from *P. inopinatus* K35. (**a**) Curacin A. (**b**) Pediocin A.

**Table 1 antibiotics-12-00676-t001:** LAB strains known to be effective against *P. aeruginosa*.

No.	Source	*LAB Strain Name*	Inhibition	Citation
1	Oral cavity	*Limosilactobacillus fermentum* ES.A.2	95%	[12]
2		*Limosilactobacillus fermentum* ES.F.115	93%	
3		*Limosilactobacillus fermentum* ES.A.6a	88%	
4		*Limosilactobacillus fermentum* ES.A.1a	88%	
5	Milk and yogurt	*Limosilactobacillus fermentum* L1	100%	[6]
6		*Limosilactobacillus fermentum* L2	100%	
7	Milk, yogurt, buttermilk, idli batter	*Levilactobacillus brevis* SII	52.63%	[13]
8	Sauerkraut	*Lactiplantibacillus plantarum* ATCC 10241	100%	[14]
9	Human	*Lactobacillus acidophilus* ATCC 4356^T^	99.99%	[15]

**Table 2 antibiotics-12-00676-t002:** Antimicrobial activity as diameter of inhibition zone of *P. inopinatus* K35 using disc diffusion assay. Data are presented as mean ± SD of the results from three replicates. * ND, non-detected.

Strain		Zone of Inhibition (mm)		
		*P. aeruginosa* KCTC 2513	*P. aeruginosa* CCARM 0223	*P. aeruginosa* CCARM 0224
LAB supernatant (100 µL/disc)				
	*P. inopinatus* K35	17.5 ± 0.87	17.33 ± 1.26	18.5 ± 0.50
Antibiotic				
	Ampicillin (5 mg/disc)	20.17 ± 1.04	16.67 ± 0.58	ND
	Gentamicin (2 µg/disc)	ND *	ND	ND

**Table 3 antibiotics-12-00676-t003:** The minimum inhibitory concentration (MIC) and minimal bactericidal concentration (MBC) of CFSs of *P. inopinatus* K35.

Strain	MIC (mg/mL)	MBC (mg/mL)
*P. aeruginosa* KCTC 2513	≥2.5	5
*P. aeruginosa* CCARM 0223	≥2.5	5
*P. aeruginosa* CCARM 0224	≥2.5	5

**Table 4 antibiotics-12-00676-t004:** Biochemical and cultural characteristics of strain K35 and *Pediococcus inopinatus* DSM 20285^T^. +, Positive; −, negative; w, weakly positive.

Characteristics	K35	DSM 20285^T^
Isolation source	Kimchi	Brewery yeast
Growth at		
25 °C	+	+
30 °C	+	+
37 °C	w	w
45 °C	w	w
Acid produced from:		
D-galactose	+	+
D-glucose	+	+
D-fructose	+	+
D-mannose	+	+
Methyl-α-D-mannopyranoside	+	−
N-acetylglucosamine	+	+
Amygdalin	w	+
Esculin ferric citrate	+	+
Salicin	+	+
D-cellobiose	+	+
D-maltose	w	w
D-trehalose	+	+
Gentiobiose	+	+
D-turanose	+	−

**Table 5 antibiotics-12-00676-t005:** Identification of antimicrobial compounds from *P. inopinatus* K35.

No.	Compound	Observed RT (min)	Formula	Theoretical Mass (*m/z*)	Observed Mass (*m/z*)	Mass Error (mDa)	Adducts
1	Curacin A	2.61	C_23_H_35_NOS	372.2367	372.2344	−2.2	-H
2	Pediocin A	3.37	C_46_H_68_N_12_O_14_S	1043.4626	1043.4577	−4.9	-H

## Data Availability

The data presented in this study are available in this paper.

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
