# Peer review of "Antimicrobial and Antibiofilm Effect of Bacteriocin-Producing Pediococcus inopinatus K35 Isolated from Kimchi against Multidrug-Resistant Pseudomonas aeruginosa"

_antibiotics, 2023, doi:10.3390/antibiotics12040676_

Round 1

Reviewer 1 Report

Line 66-67: Please rephrase

Line 79: Please add molecular characterization

line 91: add caption to the table. write correctly the µg instead of ug

line 116: correct here and overall scientific names, i.e. should be italic

line 136: full name for TIC

Figure 5: Please add details to every pictures

Please elaborate the discussion more.

Author Response

We highly appreciate the reviewer’s constructive and helpful comments on our manuscript. As suggested by the reviewer, we have carefully responded (marked in blue) to address the reviewer’s comments and revised manuscript (marked in red). We hope that the reviewer will find our responses to the comments satisfactory.

Reviewer 1:

Comments:

Line 66-67: Please rephrase

â–¶ According to the reviewer's opinion, the sentence has been revised.

Lines 71–74: Pediococcus inopinatus is a gram-positive lactic acid bacteria species, belonging to the genus Pediococcus and the family Lactobacillaceae, which can exist naturally in plants and fruits and is frequently connected to breweries or other settings where alcoholic beverages are made [12], including wine [13].

Line 79: Please add molecular characterization

â–¶ According to the reviewer's opinion, the sentence has been revised.

Lines 95-100: Strains were isolated from Kimchi using the color-changing characteristic of bromocresol purple (BCP) from purple to yellow when the pH is lowered by LAB. In this study, 40 strains that form yellow rings around colonies in BCP agar (Eiken Chemical Co., Ltd., Tokyo, Japan) plate were selected and used for antimicrobial screening. As a result of antimicrobial screening of the selected 40 strains, according to the disc diffusion assay, strain K35 with antimicrobial activity against P. aeruginosa was selected.

Line 91: add caption to the table. write correctly the µg instead of ug

â–¶ ug was corrected to µg, and added a caption to Table 2.

Lines 109–110: Antimicrobial activity as diameter of inhibition zone of P. inopinatus K35 using disc diffusion assay. Data are presented as mean ± SD of the results in three replicates. *ND, non-detected.

line 116: correct here and overall scientific names, i.e. should be italic

â–¶ All scientific names not written in italics have been corrected.

Lines 137, 139, 142, 145, 186, 195, 196: P. aeruginosa

Lines 188, 194, 195: P. inopinatus

line 136: full name for TIC

â–¶ According to the reviewer's opinion, the sentence has been revised.

Lines 167–168: The total ion chromatogram (TIC) and UV chromatograms of the K35 culture broth are shown in Figure 6.

Figure 5: Please add details to every pictures

â–¶ According to the reviewer's opinion, added details on figure 5.

Figure 5. Scanning electron microscopy (SEM) (magnification, 50,000 ×, bar 1µm) images of P. aeruginosa. (a) Untreated P. aeruginosa KCTC 2513 cells. (b) Untreated P. aeruginosa CCARM 0223 cells. (c) Untreated P. aeruginosa CCARM 0224 cells. (d) P. aeruginosa KCTC 2513 cells treated with 5mg/mL of P. inopinatus K35 CFS for 24 hours. (e) P. aeruginosa CCARM 0223 cells treated with 5mg/mL of P. inopinatus K35 CFS for 24 hours. (f) P. aeruginosa CCARM 0224 cells treated with 5mg/mL of P. inopinatus K35 CFS for 24 hours.

Please elaborate the discussion more.

â–¶ According to the reviewer's opinion, the discussion was entirely revised.

Lines: 181-244

Reviewer 2 Report

Yi and Kim report a study on isolating antibacterial compounds from Kimchi. They focus on antibiotic resistance P. aeruginosa and isolate a bacterial strain that produces two compounds that they identify as Curacin A and Pediocin A. Both of these are known compounds and this study therefore did not discover new antibiotic lead compounds. Some discussion of what is known about these compounds would have improved this manuscript.

English language issues:

L 66: incomplete sentence (‘of Lactobacillaceae. may naturally..’)

L 164: this is a convoluted sentence that should be broken into two sentences.

L 195: this is not a correct English sentence. A second subject is missing.

L 206: incomplete sentence.

L236: the solution was discarded and the solution was washed?? should probably mean the solution was discarded and the well were washed

Author Response

We highly appreciate the reviewer’s constructive and helpful comments on our manuscript. As suggested by the reviewer, we have carefully responded (marked in blue) to address the reviewer’s comments and revised manuscript (marked in red). We hope that the reviewer will find our responses to the comments satisfactory.

Reviewer 2:

Yi and Kim report a study on isolating antibacterial compounds from Kimchi. They focus on antibiotic resistance P. aeruginosa and isolate a bacterial strain that produces two compounds that they identify as Curacin A and Pediocin A. Both of these are known compounds and this study therefore did not discover new antibiotic lead compounds. Some discussion of what is known about these compounds would have improved this manuscript.

English language issues:

L 66: incomplete sentence (‘of Lactobacillaceae. may naturally..’)

â–¶ According to the reviewer's opinion, the sentence has been revised.

Lines 71–74: Pediococcus inopinatus is a gram-positive lactic acid bacteria species, belonging to the genus Pediococcus and the family Lactobacillaceae, which can exist naturally in plants and fruits and is frequently connected to breweries or other settings where alcoholic beverages are made [12], including wine [13].

L 164: this is a convoluted sentence that should be broken into two sentences.

â–¶ According to the reviewer's opinion, the sentence has been revised.

Lines 193–202: In previous studies reported that antibiotic-resistant strains have the tendency to enhance biofilm formation in these strains, making it harder to antibiotic to penetrate [15]. This biofilm strengthening is consistent with the results of this study, in which antibiotic-resistant strains P. aeruginosa CCARM 0223 and P. aeruginosa CCARM0224 produced more biofilms than normal strain P. aeruginosa KCTC 2513 (Figure 4).

Lines 209–214: A recent research stated that Antibiotic-resistant strains were observed to have cell elongation than normal bacteria [15]. According to another study reported that wherein antibiotics such as erythromycin and clarithromycin inhibit protein synthesis of P. aeruginosa and eventually losses the cell viability [22]. The shrinkage of the P. aeruginosa cell by treated sample in our findings can be attributed the possible ability by CFS of K35 to inhibit protein synthesis (Figure 5).

L 195: this is not a correct English sentence. A second subject is missing.  

â–¶ According to the reviewer's opinion, the sentence has been revised.

Lines 253–260: The Kimchi soup was serially diluted in 0.85% (w/v) saline solution to a concentration of 10-6 to 10-9, and were spread onto BCP agar. Then the plates were incubated for 2 days at 30 °C.

L 206: incomplete sentence.

â–¶ According to the reviewer's opinion, the sentence has been revised.

Lines 268–269: For biochemical analysis, a carbohydrate fermentation test was performed using an analytical profile index (API) 50 CH kit (BioMérieux, Marcy l'Etoile, France) [26].

L236: the solution was discarded and the solution was washed?? should probably mean the solution was discarded and the well were washed

â–¶ According to the reviewer's opinion, the sentence has been revised.

Lines 307-308: After incubation, the solution was discarded, and each well was washed with phosphate-buffered saline (PBS) twice.

Reviewer 3 Report

The manuscript entitled “Antimicrobial and antibiofilm effect of bacteriocin-producing Pediococcus inopinatus K35 isolated from Kimchi against multidrug-resistant Pseudomonas aeruginosa” is a study of antibacterial ant antibiofilm effects of strain K35 isolated form LAB that was obtained from Kimchi. Antibacterial testing was performed on the P. aeruginosa strain. However, prior to the publication of this manuscript, some huge changes must be made. So, at this stage I’m recommending to the authors to take more time to prepare the manuscript till next submission.

General conclusion is that whole manuscript should be rewritten, language is poor and very hard to read. Figures are not cited correctly and they don’t appear correctly in the manuscript. All figures and tables should appear when they are firstly cited in text. Also, the results should be presented more clearly, and the material and method part should contain more information. In conclusion changes have to be made before considering the publication. Below authors can find some of the specific comments:

Abstract:

Line 14 and 15: “Antibacterial activity and antimicrobial film formation experiments…” this sentence should be rewritten as well.  It is enough to use one term antibacterial or antimicrobial. But since the paper is about testing the resistance of Pseudomonas aeruginosa (bacterial strain) the term antibacterial activity can be used.

Line 19: “MRPA” abbreviation should be explained.

Missing spaces before brackets.

Introduction:

Line 36: references should be in the one bracket.

Line 40: pay attention on the words that are italic. Revise all manuscript accordingly.

Line 67: does the sentence ends or continues?

Results:

Line 79: “BCP” abbreviation should be explained.

Lines 79-81: My question is did you isolate 40 strains or just one that was tested (K35)? This is not clear out of this sentence.

Spaces needed before units. Revise whole manuscript.

Line 102: “alpha” should be “α”

Line 112: “Figure 3”. Probably typing mistake, but I’m sure that result of the inhibition of biofilm formation are not displayed in Figure 3.

Figures and tables should appear in after the text they are mentioned in.

Line 125 and Line 136: Again, the wrong Figure has been citated in the text.

Line 178: “Figure 6. LC-QTOF-MS chromatogram of cell-free supernatant from P. inopinatus K35. (a) Curacin A. (b) Pediocin A.” Authors should in supplementary file add MS spectrum of identified components with few fragmentation ions explained.

Discussion:

All discussion part should be rewritten, english language is poor and hard to understand.

Conclusion part of the manuscript is missing.

Materials and methods:

Line 214: Where is the end of the sentence?

Part 4.3. Disc diffusion assay- the measuring of the diameter of the inhibitory zone should be explained.

Part 4.4. Broth microdilution method- how did you calculate MIC and MBC values? Results are obtained by using a microplate reader, but MIC and MBC values are probably obtained on different way. Which concentrations of the selected LAB were used in this method?

Part 4.5. Biofilm formation inhibitions Assay: in which units are the concentrations used for this assay? Here also has to be added the method for obtaining the presented results.

Author Response

We highly appreciate the reviewer’s constructive and helpful comments on our manuscript. As suggested by the reviewer, we have carefully responded (marked in blue) to address the reviewer’s comments and revised manuscript (marked in red). We hope that the reviewer will find our responses to the comments satisfactory.

Reviewer 3:

The manuscript entitled “Antimicrobial and antibiofilm effect of bacteriocin-producing Pediococcus inopinatus K35 isolated from Kimchi against multidrug-resistant Pseudomonas aeruginosa” is a study of antibacterial ant antibiofilm effects of strain K35 isolated form LAB that was obtained from Kimchi. Antibacterial testing was performed on the P. aeruginosa strain. However, prior to the publication of this manuscript, some huge changes must be made. So, at this stage I’m recommending to the authors to take more time to prepare the manuscript till next submission.

General conclusion is that whole manuscript should be rewritten, language is poor and very hard to read. Figures are not cited correctly and they don’t appear correctly in the manuscript. All figures and tables should appear when they are firstly cited in text. Also, the results should be presented more clearly, and the material and method part should contain more information. In conclusion changes have to be made before considering the publication. Below authors can find some of the specific comments:

Abstract:

Line 14 and 15: “Antibacterial activity and antimicrobial film formation experiments…” this sentence should be rewritten as well.  It is enough to use one term antibacterial or antimicrobial. But since the paper is about testing the resistance of Pseudomonas aeruginosa (bacterial strain) the term antibacterial activity can be used.

â–¶ According to the reviewer's opinion, the sentence has been revised. In addition, all antimicrobial or antibacterial terms written in the manuscript were unified as antimicrobial.

Lines 14–16: Antimicrobial activity and antibiofilm activity were performed against multidrug-resistant (MDR) Pseudomonas aeruginosa using cell-free supernatants of lactic acid bacteria (LAB) isolated from Kimchi.

Line 19: “MRPA” abbreviation should be explained. Missing spaces before brackets.

â–¶ According to the reviewer's opinion, the sentence has been revised.

Lines 18–19: Cell-free supernatants (CFS) of strain K35 isolated from Kimchi effectively inhibited the growth of MDR P. aeruginosa.

â–¶ Spaces have been rechecked and corrected.

Lines 15–16: multidrug-resistant (MDR) Pseudomonas aeruginosa

Line 16: lactic acid bacteria (LAB)

Line 18: Cell-free supernatants (CFS)

Line 169: software (UNIFI v1.8.1)

Line 204: EzBioCloud database (https://www.ezbiocloud.net/identify)

Line 205: replications [30,31].

Table 2.: Zone of inhibition (mm), Ampicillin (5 mg/disk)

Introduction:

Line 36: references should be in the one bracket.

â–¶ References were put into one bracket.

Line 35: [3,4].

Line 40: pay attention on the words that are italic. Revise all manuscript accordingly.

â–¶ According to the reviewer's opinion, it was rechecked that the italics were properly used in the manuscript, and the wrong parts were corrected.

Line 39: studied

Lines 137, 139, 142, 145, 186, 195, 196: P. aeruginosa

Lines 188, 194, 195: P. inopinatus

Line 67: does the sentence ends or continues?

â–¶ It was confirmed that a period was inserted instead of a comma, and the sentence was corrected.

Lines 71–74: Pediococcus inopinatus is a gram-positive lactic acid bacteria species, belonging to the genus Pediococcus and the family Lactobacillaceae, which can exist naturally in plants and fruits and is frequently connected to breweries or other settings where alcoholic beverages are made [12], including wine [13].

Results:

Line 79: “BCP” abbreviation should be explained.

â–¶ According to the reviewer's opinion, the sentence has been revised.

Lines 95-96: Strains were isolated from Kimchi using the color-changing characteristic of bromocresol purple (BCP) from purple to yellow when the pH is lowered by LAB.

Lines 79-81: My question is did you isolate 40 strains or just one that was tested (K35)? This is not clear out of this sentence.

â–¶ In this study, we are isolated 40 strains from Kimchi and used for antimicrobial screening. And according to the reviewer's opinion, the sentence has been revised.

Lines 96-100: In this study, 40 strains that form yellow rings around colonies in BCP agar (Eiken Chemical Co., Ltd., Tokyo, Japan) plate were selected and used for antimicrobial screening. As a result of antimicrobial screening of the selected 40 strains, according to the disc diffusion assay, strain K35 with antimicrobial activity against P. aeruginosa was selected.

Spaces needed before units. Revise whole manuscript.

â–¶ The space in front of the unit was checked again and corrected appropriately.

Lines 101: 17.5 ± 0.87, 17.33 ± 1.26 and 18.5 ± 0.50 mm

Line 102: “alpha” should be “α”

â–¶ According to the reviewer's opinion, the sentence has been revised.

Table 4: Methyl-α-D-mannopyranoside

Line 112: “Figure 3”. Probably typing mistake, but I’m sure that result of the inhibition of biofilm formation are not displayed in Figure 3.

â–¶ The figure number has been corrected.

Line 138: The result of biofilm formation inhibitory effect against P. aerusionsa are shown in Figure 4.

Figures and tables should appear in after the text they are mentioned in.

â–¶ According to the reviewer's opinion, the positions of figures and tables have been properly rearranged.

Line 125 and Line 136: Again, the wrong Figure has been citated in the text.

â–¶ The figure numbers have been corrected.

Lines 153–155: The morphological changes of P. aeruginosa KCTC 2513, P. aeruginosa CCARM 0223 and P. aeruginosa CCARM 0224 with MBC of CFS from K35 were visualized by Scanning electron microscopy (SEM) (Figure 5).

Lines 167-168: The total ion chromatogram (TIC) and UV chromatograms of the K35 culture broth are shown in Figure 6.

Line 178: “Figure 6. LC-QTOF-MS chromatogram of cell-free supernatant from P. inopinatus K35. (a) Curacin A. (b) Pediocin A.” Authors should in supplementary file add MS spectrum of identified components with few fragmentation ions explained.

â–¶ According to the reviewer's opinion, supplemental Figure 2 provided.

Supplementary Fig. 2. Identified fragment ionization of cell-free supernatant from P. inopinatus K35. (a) Full MS spectrum of curacin A. (b) fragment MS spectrum with curacin A fragment being marked as yellow. (c) identified fragment ion list of curacin A. (d) full MS spectrum of pediocin A. (e) fragment MS spectrum with pediocin A fragment being marked as yellow. (f) identified fragment ion list of pediocin A.

Discussion:

All discussion part should be rewritten, english language is poor and hard to understand.

â–¶ According to the reviewer's opinion, the discussion was entirely revised.

Lines 181–207\\44

Conclusion part of the manuscript is missing.

â–¶According to the reviewer’s opinion, the conclusion part added.

Lines 247–255:

Line 214: Where is the end of the sentence?

â–¶ A period was added.

Lines 275–276: Then, the selected LAB was grown in MRS broth for 48 h at 30 °C to produce a mass amount of metabolite.

Part 4.3. Disc diffusion assay- the measuring of the diameter of the inhibitory zone should be explained.

â–¶ According to the reviewer's opinion, the sentence has been revised.

Lines 286-288: The antimicrobial effects of LAB were analyzed by measuring the diameter of the inhibitory zone in millimeters using calipers [34].

Part 4.4. Broth microdilution method- how did you calculate MIC and MBC values? Results are obtained by using a microplate reader, but MIC and MBC values are probably obtained on different way. Which concentrations of the selected LAB were used in this method?

â–¶ 96-well microplate dilution method was described in the Clinical and Laboratory Standards Institute guidelines. According to the reviewer's opinion, written in more detail.

Lines 290-301: The broth microdilution method described in the Clinical and Laboratory Standards Institute guidelines was utilized to confirm the MIC and MBC of selected LAB.                                                        A 96-well microtiter plate (Thermo Fisher Scientific, Waltham, MA, USA) was used to inoculate serially diluted evaporated-CFS of selected LAB. To each well containing 100 μL of serially diluted evaporated-CFS, 100 μL of P. aeruginosa strains in NB (1×106 CFU/mL) was added, and the plates were incubated at 30 °C for 24 h. Then, the optical density (OD) was measured at 595 nm, and the concentration showing OD of 20% or less compared to the control group was set as MIC by a microplate reader (Molecular Devices, San Francisco, CA, USA). After that, a portion of each well was streaked on a NB agar plate using a 1uL inoculation loop. The plates were incubated at 37 °C for 24 h to observe the number of colonies. The least concentration without colonies grown on the NB agar plates was recorded as the MBC.

Part 4.5. Biofilm formation inhibitions Assay: in which units are the concentrations used for this assay? Here also has to be added the method for obtaining the presented results.

â–¶ The units used in the experiment were written.

Lines 304-307: To determine whether treatment of evaporated-CFS reduces the amount of biofilm, 100 μL each of dilution of P. aeruginosa strains (1×106 CFU/mL) and different concentrations of CFS-K35 (0, 0.31, 0.63, 1.25, 2.5 and 5 mg/mL) were filled into 96-well microtiter plates and incubated for 24 h at 37 °C without shaking.

â–¶ According to the reviewer's opinion, we added the method for obtaining the presented results.

Lines 276–280: Culture media was then centrifuged for 10 min at 12,000 rpm, and the supernatant was filtered using a 0.22 μm membrane filter. Subsequently, the filtrates were used directly as samples for antimicrobial activity screening, and concentrated filtrates using a centrifugal evaporator (EYELA, Tokyo, Japan) at 45 °C were utilized for further antimicrobial and antibiofilm tests.

Round 2

Reviewer 3 Report

The authors have introduced all changes suggested and the manuscript under title “Antimicrobial and antibiofilm effect of bacteriocin-producing Pediococcus inopinatus K35 isolated from Kimchi against multidrug-resistant Pseudomonas aeruginosa” has now been significantly improved. I would only suggest ones more, language style correction, it the new parts of the manuscript that have been introduced in the original version.